# "I'd rather wait and see what's around the corner": A multi-perspective qualitative study of treatment escalation planning in frailty

Adam Lound[1]*, Jane Bruton[1], Kathryn Jones[1], Nira Shah[2], Barry Williams[2], Jamie Gross[3], Benjamin Post[4,5], Sophie Day[1], Stephen J. Brett[6,7], Helen Ward[1,7,8]

1 Patient Experience Research Centre, School of Public Health, Imperial College London, London, United Kingdom, 2 ProsPECT Study Public Advisor, Patient Experience Research Centre, School of Public Health, Imperial College London, London, United Kingdom, 3 London North West University Healthcare NHS Trust, London, United Kingdom, 4 Department of Bioengineering and Department of Computing, Imperial College London, London, United Kingdom, 5 UKRI Centre in AI for Healthcare, Imperial College London, London, United Kingdom, 6 Department of Surgery and Cancer, Imperial College London, London, United Kingdom, 7 Imperial College Healthcare NHS Trust, London, United Kingdom, 8 National Institute for Health Research Imperial Biomedical Research Centre, London, United Kingdom

* a.lound@imperial.ac.uk

**Data Availability Statement:** This is a qualitative study and the full study data cannot be made publicly available due to the sensitive nature of this

## Abstract

### Introduction

People living with frailty risk adverse outcomes following even minor illnesses. Admission to hospital or the intensive care unit is associated with potentially burdensome interventions and poor outcomes. Decision-making during an emergency is fraught with complexity and potential for conflict between patients, carers and clinicians. Advance care planning is a process of shared decision-making which aims to ensure patients are treated in line with their wishes. However, planning for future care is challenging and those living with frailty are rarely given the opportunity to discuss their preferences. The aim of the ProsPECT (Prospective Planning for Escalation of Care and Treatment) study was to explore perspectives on planning for treatment escalation in the context of frailty. We spoke to people living with frailty, their carers and clinicians across primary and secondary care.

### Methods

In-depth online or telephone interviews and online focus groups. The topic guide explored frailty, acute decision-making and planning for the future. Data were thematically analysed using the Framework Method. Preliminary findings were presented to a sample of study participants for feedback in two online workshops.

### Results

We spoke to 44 participants (9 patients, 11 carers and 24 clinicians). Four main themes were identified: frailty is absent from treatment escalation discussions, planning for an uncertain future, escalation in an acute crisis is *'the path of least resistance'*, and approaches to facilitating treatment escalation planning in frailty.

data, the agreements and procedures governing usage and sharing of the collected dataset established by the research institution (Imperial College London) and Research Ethics Board (21/LO/0125). However, we welcome approaches to the Patient Experience Research Centre from bona fide researchers who may wish to explore the data further in collaboration with our team. Interested parties should provide a data access request to patientexperience@imperial.ac.uk and following approval of the request, establish a data-sharing agreement between the investigators and the Study PIs.

**Funding:** This project was funded by two grants, HW received funding from the NIHR Imperial Biomedical Research Centre (NIHR203323) https://imperialbrc.nihr.ac.uk/. SJB received funding from the Imperial College Healthcare Trust Research Capability Fund (WSAA_P83429) https://www.imperial.nhs.uk/. The funders had no role in study design, data collection and analysis, decision to publish, or preparation of the manuscript. HW is an NIHR Senior Investigator and acknowledges support from the NIHR ARC Northwest London and the NIHR School of Public Health Research. BP receives general support from the UKRI CDT in AI for Healthcare (EP/S023283/1) http://ai4health.io.

**Competing interests:** The authors have declared that no competing interests exist.

## Conclusion

Barriers to treatment escalation planning include a lack of shared understanding of frailty and uncertainty about the future. Emergency decision-making is focussed on survival or risk aversion and patient preferences are rarely considered. To improve planning discussions, we recommend frailty training for non-specialist clinicians, multi-disciplinary support, collaborative working between patients, carers and clinicians as well as broader public engagement.

## Introduction

In clinical practice frailty is seen as a multidimensional, age-related syndrome characterised by a loss of in-built reserves and vulnerability to stressors, estimated to affect around 11% of community-dwelling older adults worldwide [1]. Following even minor illnesses frail individuals risk adverse outcomes including functional impairment, hospitalisation and mortality [2]. Hospital or intensive care unit (ICU) admission is associated with prolonged lengths of stay, complications such as delirium, and mortality [3, 4]. Survival following a hospital stay is correlated with disability, poor quality of life and increasing care needs [3, 5].

Decision-making during an emergency presents a significant challenge. Patients may lack capacity and there is potential for conflict between surrogate decision-makers [6]. Treatment escalation planning involves prospective decision-making regarding the range of increasingly intensive or complex interventions offered during an admission. Advance care planning (ACP) promotes shared decision-making, enabling patients to make informed choices about their future healthcare, including treatment escalation and end of life care [7, 8]. However, ACP is poorly implemented in frailty with patients rarely given the chance to express their preferences [9, 10]. Emergency care and treatment plans, which summarise patient goals of care for emergency situations, have emerged to support acute decision-making but concerns about their use in practice include a tendency to focus solely on resuscitation decisions or end of life planning [7, 8].

Identifying and managing frailty is complex due to varying diagnostic definitions, different clinical disease models and a multitude of available screening and severity scoring instruments [11]. There has been growing interest in the role of electronic health records to support proactive frailty identification and management at a population level. In 2017 the UK introduced primary care screening using the electronic frailty index to classify severity and provide an insight into risk of hospitalisation and mortality [12]. However, communicating a diagnosis of frailty is challenging because the term is widely perceived to be negative, stigmatising and potentially discriminatory.

Our study was developed alongside an Imperial College project identifying community-dwelling patients at risk of ICU admission using artificial intelligence and machine-learning approaches to analyse routine healthcare data. We initially planned to explore perspectives of patients and their carers once identified as at risk of ICU admission. In discussion with clinical and public partners, it was recognised that we needed to expand beyond ICU admission decisions to capture a broader spectrum of treatment escalation dilemmas faced by patients, carers and clinicians.

The aim of the ProsPECT (Prospective Planning for Escalation of Care and Treatment) study was to explore participant perspectives on planning for treatment escalation in the context of frailty. We drew on the experiences and perspectives of those living with frailty, their

carers and clinicians to explore the concept of frailty and decision-making regarding the potential risks, benefits and alternatives to treatment escalation.

## Methods

### Study design

We undertook in-depth interviews (April 2021- February 2022) and focus groups (June 2022). The study received ethical approval from the London Chelsea Research Ethics Committee (reference 21/LO/0125, March 2021) and the Health Research Authority.

### Sampling and recruitment of participants

We recruited people living with frailty using purposive sampling for diversity of demographic characteristics, level of frailty and acute healthcare experience. Participants were suitable if they were assessed as being frail by the referring clinician. We recruited from three chronic disease clinical pathways with a high prevalence of frailty (respiratory, renal and heart failure) and from clinical databases of COVID-19 survivors from two London ICU's [13–15]. Potential participants were approached by clinicians and with their agreement referred to the research team with contact details, a brief summary of their medical history and a Clinical Frailty Score (CFS). The CFS is a widely used screening tool to identify and score severity [16].

Carers were either approached by clinicians or nominated by patient participants. For this study 'carer' refers to an unpaid family member, spouse or a significant other who provides care for a person living with frailty and supports their healthcare decision-making [17]. We sought diversity of demographic characteristics and experience in healthcare decision-making.

We recruited primary and secondary care clinicians who were involved in managing frailty, again purposive sampling was used for diversity in demographic characteristics and professional role. Clinicians were recruited through professional networks, special-interest groups and general practitioner meetings. Inclusion criteria for all participants can be found in Table 1.

**Table 1. Inclusion criteria.**

|  | Inclusion Criteria |
|---|---|
| People living with frailty | Aged 18 years or older<br>Living with frailty–confirmed using Rockwood Clinical Frailty Score (CFS)[1]<br>Able to read and understand the participant information sheet (PIS)<br>Able to give informed written or witnessed verbal consent[2.]<br>Able to speak and understand conversational English |
| Carers | • Aged 18 years or older<br>• Adults who would potentially make decisions with or on behalf of someone living with frailty (e.g. family member, partner, carer, friend)[3]<br>• Able to read and understand the PIS<br>• Able to give informed written consent<br>• Able to speak conversational English |
| Clinicians | • Clinicians with experience working with those living with long-term conditions and frailty<br>• Has read the PIS and given informed written consent |

[1] The CFS provides a severity score for frailty (0–9), those scoring 5 (mildly frail) or above based on referring clinician assessment were eligible for inclusion.

[2] Confirmed by referring clinician.

[3] Carers of individuals who had the capacity to make informed decisions were eligible for inclusion.

All participants were contacted via email or telephone and provided with a participant information sheet (PIS). AL obtained informed, written consent and demographic information using paper or online forms. The sensitive nature of the study was reiterated and verbal consent to record was obtained at the start of interviews and focus groups. To protect participant anonymity AL assigned serial numbers to participants. Personal information and contact details were stored securely and separately from research data and in line with the host institutions governance requirements, as were electronic files linking serial numbers to participants. Other authors did not access this information.

## Data collection

Participants were given the choice of telephone or online interview. No face-to-face participation was offered due to COVID-19 restrictions. Two clinician focus groups, one involving doctors and one involving nurses and allied health professionals took place online.

Draft topic guides, based on existing literature, were refined through feedback from our public partners and adapted for each stakeholder group. The topic guide explored: the concept of frailty, acute decision-making and planning for a future crisis. All interviews were conducted by a research physiotherapist (AL). The focus groups were co-facilitated by AL and the research team (SJB and/or JG; JB, KJ). Interviews and focus groups were recorded transcribed and anonymised prior to analysis by an external provider UK Transcription, https://www.uktranscription.com/ with whom Imperial have a confidentiality agreement.

## Analysis

Transcripts were uploaded and managed using NVivo (QSR International (UK) Ltd) and analysed using the Framework Method [18]. The Framework Method is a systematic approach to qualitative analysis which includes the use of a matrix to organise and summarise data. Familiarisation and coding were undertaken by two researchers (AL and KJ) with 19 of the 25 transcripts double coded. Codes were then indexed and charted (AL) into a matrix based on the key areas explored in the topic guides: frailty, acute crisis and prospective planning. One researcher (AL) created a matrix for each stakeholder group and a separate matrix for the focus groups. Three research team members (AL, KJ, JB) met regularly to iteratively develop the analytic framework and clarify themes and interpretation of the findings. Findings were presented to the wider research team and public advisors for review.

## Patient and public involvement and reflexivity

A steering group (including public partners, clinicians and commissioners) met throughout the study. All study documents were developed with feedback from our public partners. The research team includes clinicians with experience of caring for or clinically managing people living with frailty. AL is a research physiotherapist with training in qualitative methods. Participants interviewed by AL were made aware of his clinical background; three participants were former patients of AL but were interviewed about their role as carers. JB, KJ, SJB, JG and HW are experienced clinical researchers with training or experience in using qualitative methods. We were aware that our backgrounds might influence study design, analysis and interpretation of findings, and sought to move beyond individual, potentially biased, perspectives by ensuring analysis was undertaken in liaison with public partners to agree findings. Initial findings were presented to a purposive sample of study participants in two online workshops—providing a forum to interrogate the analysis and discuss dissemination of findings.

## Results

We conducted 25 interviews with 30 participants (nine patients, 11 carers and 10 clinicians) and two focus groups with 14 clinicians. The focus groups and 23 of the interviews were online (MS Teams or Zoom) and two interviews were by phone. Four of the interviews involved both patient and carer and one interview involved two clinicians. To our knowledge, two participants have died since their interviews. Demographic details of study participants can be found in Table 2. Given the complexity of the topics covered in our study and the wide range of participant backgrounds we did not reach data saturation.

We identified four main themes: Frailty is absent from treatment escalation discussions, planning for an uncertain future, treatment escalation in an acute crisis is *'the path of least resistance'*, approaches to facilitating treatment escalation planning in frailty. Themes and subthemes can be found in Fig 1.

## Theme one: Frailty is absent from treatment escalation discussions

### Patients and carers: *'We don't accept that'*

People living with frailty described finding themselves in a *'bad state'* following an insidious physical and functional decline and were unable to disentangle the role that chronic disease progression, slow recovery from infections, medication interactions and ageing played in their changing health. Patients were able to name and describe their long-term conditions, but none used the term frailty. When it was introduced (without definition) by the researcher, some patients recognised the term as a marker of physical or cognitive decline associated with ageing–*'I don't stride out on the pavement with confidence. I'm not strong anymore, so I presume that's what frail means. . .'.* However, they did not recognise the link between their current health state and increased vulnerability, and most felt that frailty did not apply to them or rejected the term as it *'symbolises that you've already arrived at the end'.* Carers recognised that their loved ones had a *'risky outlook'* and were constantly watching for potential problems such as falls or infections. Although they understood the term frailty as a marker of declining

**Table 2. Participant demographics.**

| Clinicians (n = 24) | Patients (n = 9) | Carers (n = 11) |
|---|---|---|
| **Age:**<br>Median 42 (26–58) | **Age:**<br>Median 71 (63–84) | **Age:**<br>Median 60 (36–83) |
| **Gender:**<br>Male: 5<br>Female: 19 | **Gender:**<br>Male: 5<br>Female: 4 | **Gender:**<br>Male: 1<br>Female: 9<br>Undisclosed: 1 |
| **Ethnicity:**<br>White: 15<br>Asian: 7<br>Black: 1<br>Mixed: 1 | **Ethnicity:**<br>White: 7<br>Asian: 1<br>Mixed: 1 | **Ethnicity:**<br>White: 7<br>Black: 1<br>Asian: 2<br>Mixed: 1 |
| **Profession:**<br>Doctors: 11<br>Nurses: 8<br>Allied Health Professionals: 5 | **Clinical Frailty Score (CFS)[1]:**<br>CFS 5: 2<br>CFS 6: 6<br>CFS 7: 1<br>**Admission to intensive care unit:** 4 | **Relationship:**<br>Wife: 6<br>Husband: 1<br>Daughter: 3<br>Son: 1 |

[1] The CFS 5 = mildly frail, 6 = Moderately frail, 7 = Severely frail

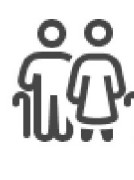
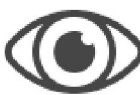
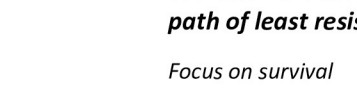
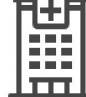
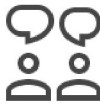

**Prospective Planning for Escalation of Care and Treatment in Frailty**

**1: Frailty is absent from treatment escalation discussions**

*Patients and Carers: 'We don't accept that'*

*Clinicians: Lacking confidence and 'skirting around the issue'*

**2: Planning for an uncertain future**

*Patient and carer uncertainty: 'I'd rather wait and see what's around the corner'*

*Clinician uncertainty: 'A fantastically complex scenario'*

*Clashing perspectives: Potential for conflict when making planning for treatment escalation*

**3: Treatment escalation in an acute crisis is *'the path of least resistance'***

*Focus on survival*

*Risk Aversion leads clinicians to 'err on the side of doing everything'*

**4: Approaches to facilitating treatment escalation planning in frailty**

*Content: Fostering a shared understanding*

*Context: Timing, trust and support*

**Fig 1. Summary of themes and subthemes.**

health and increased vulnerability, they rejected it as a diagnostic label as they felt it was synonymous with negative portrayals of ageing or associated with *'giving up.'*

> *"We don't accept that, no, no, no. . .the word frailty I rarely use. Being weaker, yes. Being not so strong, yes. Being unwell, yes but frailty, no."* (Eileen, Carer)

## Clinicians: Lacking confidence and *'skirting around the issue'*

Many clinicians lacked confidence in frailty management. It was usually subjectively recognised as *'something is not quite right'* and patients were often overlooked until they were *'obviously frail'* leading to reactive rather than proactive management.

Clinicians expressed ambivalence about frailty screening or scoring tools due to lack of training or the perception that they did not *'tally'* with how patients presented. Clinicians described being unsure what action to take upon recognising frailty so continued to focus on

chronic disease management. Identifying frailty did not commonly result in an individualised assessment, severity scoring, documentation or changes in management nor did it provide a trigger for treatment escalation planning.

> "It's actually not something that I use or as a team we use as a marker. Although I may see it on patients' notes, or I may just look at someone and think, "Gosh you are frail." I don't put that in any kind of framework..." (HCP120, Specialist Nurse)

Communicating the concept of frailty was fraught with difficulty, some clinicians found it *'hard to even describe what frailty is'* and in some cases discussed how it *'spoilt'* established relationships–*'it's not a diagnosis that patients want to hear'*. Opportunities to provide education on the *'medical concept of frailty'* were evaded due to its associated stigma and led treatment escalation planning to be marked by an *'absence of language'*. Some clinicians described *'skirting around the issue,'* using vague or euphemistic language in which they hoped the concept of vulnerability and risk would be understood.

> "It's just an all-over check-up for you and to make sure you're managing okay...I think they know by inference that that means that we think they're a bit vulnerable, perhaps, or in more need than other patients...but we probably wouldn't say, "Frail" to them, because I don't know, that gives a horrible picture sometimes". (HCP002, GP)

However, clinicians with specialist roles or experience in frailty care were more confident in managing the condition. They proactively screened for frailty and shared the diagnosis with colleagues, patients and carers. Diagnosing frailty enabled clinicians to take a more *'holistic'* view and to *'stop and consider'* their management approach. Objective tools (e.g. CFS) helped to identify vulnerable patients and were completed with patients and carers, providing a *'relatable context'* in which to discuss frailty and the need for escalation planning. Establishing a shared understanding of frailty and vulnerability prior to introducing escalation planning provided an urgency and relevance to these conversations–*'often it is enough for them to say, "Okay, things aren't looking good, are they? I need to think"'.'*

## Theme two: Planning for an uncertain future

### Patient and carer uncertainty: *'I'd rather wait and see what's around the corner'*

Most patients had not discussed or made formal plans regarding treatment escalation as they felt that the future was unknowable—involving many possibilities, which would require decisions to be taken *'in the moment'*.

> "Possibly, I'm being silly, but I think I'd rather wait and see what's around the corner than think, "Oh, this could happen or that could happen..." Everything depends on what happens and when it happens. Then, I feel that I could deal with it." (Nina, Patient)

Patients focussed on taking a *'day to day'* approach to their declining health whereas carers were on constant alert for signs and symptoms of a crisis. When patients made formal plans or expressed opinions about the future, these related largely to end of life care rather than treatment escalation. Even when patients understood they were nearing the end of their lives and had palliative care plans in place, they did not recognise that the next acute health crisis could

be a terminal event. After describing a plan for dying at home with hospice support, one patient shared their rationale for going to hospital if they became unwell:

> *"To stabilise me, basically. . .I don't think you go into hospital with an attack like this and never leave, if you see what I mean. Or that is really towards the end."* (Jack, Patient)

## Clinician uncertainty: *'A fantastically complex scenario'*

Clinicians described treatment escalation planning for frailty as a *'fantastically complex scenario'* underpinned by multiple uncertainties including an unclear illness trajectory and prognosis, multiple treatment options and outcomes. By comparison with cancer care, there was less clarity around prognosis and clinicians felt less confident about appropriate interventions. Clinicians felt frustrated about the *'emotive'* topic of resuscitation when they considered it futile but members of the public perceived a good chance of survival. Some clinicians were restricted by the boundaries of their professional roles or care setting. For example, nurses and allied health professionals felt isolated and unable to share the burden of clinical uncertainty with the wider multidisciplinary team–*'if you do raise that, then you have to own that.'* GPs highlighted the challenge of decision-making across care settings. When patients were stable, it was difficult to engage them in escalation planning because it seemed irrelevant.

> *"The further away from the acute event people are, the more difficult it is for them to actually get their head around it."* (HCP115, GP)

## Clashing perspectives: Potential for conflict when making planning for treatment escalation

Planning for hospital admission or treatment escalation was a potential source of conflict. Many patients and carers saw hospital as a place of safety and security, but some clinicians were concerned that frail patients who were admitted—*'get worse care. It's much more invasive. . .and not consistent'*. When sharing these concerns and discussing alternatives to hospital admission, clinicians often *'clashed'* with family members who, they felt, sometimes had unrealistic expectations.

A lack of shared understanding regarding treatment escalation was evident. Patients and carers described having a *'very vague sense'* of the available options and most discussed resuscitation and ventilation but were unsure what they meant.

> *"I say resuscitation, but actually I've no idea what it means specifically or medically. I don't know what they actually do to you"*. (Sarah, Patient)

Alternatives to treatment escalation, particularly related to end of life care preferences, occasionally revealed stark differences in opinion between patients and their carers.

Participants discussed the influence of religion, culture and ethnicity in decision-making around treatment escalation. Two patients with different religious backgrounds described accepting all forms of treatment escalation and leaving it to their respective deity to decide the outcome–*'I kneel to the God'*. In some cultures, offering anything but full treatment escalation was seen as *'neglectful'* whereas, in others, some interventions were viewed negatively–*'in our culture the ventilation is dying. . .it's the last step of life'*. Different cultural attitudes towards discussions of death and dying added another layer of complexity to treatment escalation

discussions–*'the medical profession is not going to understand all the cultural nuances that exist when people talk about death'*.

## Theme three: Treatment escalation in an acute crisis is *'the path of least resistance'*

### Focus on survival

During the tumult of an acute health crisis, patients were often too unwell or lacked decision-making capacity to be involved in their own care—*'I was out of it really'*. They relied on their carers to *'do what is right'* and trusted clinicians as *'they know better'*. For carers making surrogate decisions, survival was paramount–*'do anything you can to save his life'* and alternatives to full active treatment were seen as unacceptable—*'we're not going to make that call just to let her go'*. Specific treatment options were rarely considered–*'I trust medical science, please go ahead with whatever'*- and concerns shared by doctors about the potential for poor outcomes were unexpected or actively rejected.

One carer reflected:

*"Maybe we were just like, "We don't want to hear that." You know? It was terrifying. Terrifying. These were his words–"Even if he did get through it, and I don't hold out a lot of hope, he'll never be the same man again."* (Brenda, Carer)

Some carers were reluctant to escalate treatment. One described taking a *'calculated risk'* of not calling her father's GP when he was unwell, knowing they would suggest admission to hospital where she felt he would receive poor, undignified care:

*"It's not the right time, we'll ride this out. . .I was all about maintaining his dignity as well. I thought for him, at that particular time, that was more important than trying to maintain his health"* (Maria, Carer)

### Risk aversion leads clinicians to *'err on the side of doing everything'*

During an acute health crisis there was a strong tendency towards hospital admission as the *'path of least resistance'*. With the clinical uncertainties surrounding frailty, decisions focused on reducing risk. Hospital was perceived as *'a safer place'* and alternatives to admission were much harder to organise, often caused conflict between clinician's and carers and raised concerns about deterioration–*'Will they be gasping for breath?'*. Attempts to support patients to remain at home were undermined by a lack of clinical and social services, particularly during evenings and weekends–*'out of hours is my biggest fear'*.

In hospital, clinicians described making decisions with limited information and under significant time pressure which led them to *'err on the side of doing everything'* and to focus *'very much on saving lives'*. They found it challenging to discuss stopping treatment or offering palliative care if there was *'any chance of recovery, however small'*. Although clinicians were aware of the potential for poor outcomes, some mentioned patients who had made unexpected recoveries which tempered their confidence in suggesting alternatives to escalation. One doctor reflected on the potentially negative consequences of escalation and felt that alternatives required a *'nuanced and braver decision'* which could ultimately lead to more dignified care:

*"I think, more often than not, a patient will be escalated, will go to intensive care, only to then spend four months rehabilitating and then die of pneumonia in hospital before they go. That's*

*more of a source of reflection for me, over the years, of thinking, "That was not a good death, what could we have done to make that better?"* (HCP007, Geriatrician)

Some clinicians shared positive examples of treatment escalation documentation being used to guide decisions, but many questioned their utility during an emergency. Hospital clinicians felt they often had to *'start from scratch'* as ACPs developed in the community were inaccessible, lacked the *'granularity'* needed to guide treatment options and did not account for the fact that *'people's view of what's acceptable changes quite dramatically'*. Care plans made as part of a hospital discharge often contained vague language, such as *'Hospital for reversible causes, but not for resuscitation.'* which could be interpreted in multiple ways by different community clinicians rendering them *'meaningless.'*

## Theme four: Approaches to facilitating treatment escalation planning in frailty

### Content: Fostering a shared understanding

Successful treatment escalation planning was described as a collaborative process and was likened to a *'staircase'* in which each step is an opportunity to develop a shared understanding of living with frailty, treatment escalation and patient and carer preferences.

> *"People need to understand about frailty before they can think and understand about advance care planning as patients don't realise how fragile they are"* (HCP303, Specialist Nurse)

Strategies used during conversations included the use of *'objective'* facts (e.g., survival rates following resuscitation), *'scenario roleplay'* and analogies such as *'the paper boat on the sea'*. Involving carers in planning conversations reduced the risk of conflict or misunderstandings from *'mismatched expectations'*. Clinicians sought to understand the influence of different backgrounds and experiences in decision-making and worked in partnership to develop a *'framework of what quality of life is for the patient'*. Discussions about prognosis, ceilings of treatment or end of life care required sensitivity and clarity about the rationale for different medical recommendations.

> *"This is the situation and this is what we, as the medical team, will offer, what are your thoughts, is that something you will accept?" Not having that confidence, you're more likely to say, "What do you want?" That can result, I find, in breakdowns of communication and a messy decision-making where, then, the patient doesn't quite know what the doctors are recommending."* (HCP007, Geriatrician)

### Context: Timing, trust and support

Clinicians described treatment escalation as a *'process'* that should ideally start early in the community and when patients were stable, '*You plant a seed, and then you revisit it and revisit it'*. Key moments for starting discussions were described, such as increasing care needs or following discharge from hospital --*'when people have some sense as to what that process is of escalation'*. Patients reported changing their perspectives on treatment in response to receiving information or experiencing an intervention.

When asked what information would help them make prospective decisions, most patients and carers wanted trusted clinicians to have *'honest'* conversations with them about the future.

*"I think just ask, and someone should say to you, "Well, look, personally, I think that it wouldn't really do you any good. This could happen when you come out of it, and the chances are that it's not going to be such a great outcome"* (Norman, Patient)

Trust was built through regular contact and providing practical support beyond disease management (such as advice around equipment or benefits). Long-term condition specialists were frequently cited as trusted professionals who were well-placed to lead discussions. It was more challenging for patients to develop rapport with secondary care clinicians during an acute health crisis and a lack of continuity in GP services resulted in relationships with patients sometimes being *'transitory at best'*.

Input from multidisciplinary team members enabled services to recognise frailty, develop a *'whole picture of the patient'* and support decision-making across care settings. When treatment recommendations were based on a *'consensus'* from a wider team, clinicians felt more confident in leading discussions.

## Discussion

Our study highlights the multiple complexities faced when planning for treatment escalation. Discussions are undermined by a lack of shared understanding of frailty and uncertainties about what the future holds. During an emergency, carers make surrogate decisions focussed on survival and clinicians make decisions based on averting risk, potentially limiting full consideration of patient preferences or potential consequences of escalation. We found that treatment escalation planning could be enhanced through improvements in both the context and content of planning discussions.

Our study demonstrates that the concept of frailty was largely absent from treatment escalation planning resulting in discussions being rejected, seen as irrelevant or lacking impetus. We corroborate previous research showing that frail individuals prefer to focus on their present lives and consider the term frailty to strongly connote negative perceptions of ageing [19, 20]. Negative perceptions of frailty have been shown to affect behaviour [21]. Identifying frailty lacked utility for many clinicians in our study resulting in a failure to alter clinical management or to address it during ACP discussions. Barriers to discussing frailty include time constraints, difficulty with prognostication, a lack of specific communication skills training and negative connotations of the term frailty [22, 23].

Advance care planning is recommended as a way to promote shared decision-making [24]. However, ACP is poorly implemented in the context of clinically defined frailty and in guiding emergency decision-making. Our study provides insights into challenges of the ACP process that may undermine its usefulness in acute settings. We found that ACP tended to focus on end of life care rather than discussions of treatment escalation and that documented plans were often inadequate in a crisis. A potential strategy for improvement is the Recommended Summary Plan for Emergency Care and Treatment (ReSPECT) form which has been developed to promote escalation planning conversations and documentation. However, implementation of ReSPECT, which was not widely used by clinicians in our study, has revealed similar challenges. ReSPECT was used by GPs for end of life documentation who found it challenging to make treatment recommendations outside of their speciality or across care settings [25]. This highlights the complexity of undertaking treatment escalation planning and the need for further research into both implementation and process evaluation in order to identify strategies to improve impact [10, 26].

Although planning discussions often focussed on eliciting an individual's preferences, our findings highlight that decision-making is strongly influenced by carer involvement, culture,

religion and background. Carers recognised the vulnerability of the person they were caring for, and the need to consider a future health crisis, but had rarely been involved fully in planning for the future or reported different priorities to both patients and clinicians. Patients and carers may have conflicting preferences for future care (including life-sustaining interventions) and differences which may be reduced by facilitating open dialogue during ACP [27]. During hospital admission, carers attempt to advocate for patients but find themselves playing 'second fiddle' to clinical staff and limited by organisational constraints [28]. Our findings revealed that the influence of diverse backgrounds is rarely considered during an admission which has also been reported in other studies. Ethnicity, for example, has been shown to be associated with poor access to palliative care and uptake of ACP [30].

Our study identified a training need for non-specialist clinicians undertaking treatment escalation planning in the context of frailty. There are a growing number of initiatives for frailty education and training to improve clinician confidence and competence [31]. A recently developed intervention combining training on ACP and frailty, underpinned by behaviour change theory, could be a model for improving these complex conversations [26]. Sensitively establishing a mutual understanding of frailty and the need to plan for the future requires collaboration with patients and carers. Facilitated discussions should aim to explore patient and carer perspectives, including influences such as cultural and religious background, in order to align goals of care and reduce potential conflict.

Clinician uncertainty led to planning avoidance and risk aversion during an acute crisis. Recent literature has highlighted how uncertainty is experienced across stakeholder groups and elicits different responses but strategies to address this remain poorly understood [29]. Our study identifies some potential ways to reduce uncertainty including the use of objective frailty scoring, providing multidisciplinary support for individual clinicians leading planning discussions and working collaboratively with patients and carers. However, uncertainty remains a core problem in care planning for frailty and further research is required to develop management approaches.

Care planning conversations need to start early in the community, when frailty is less severe, and develop into a 'process' of repeated discussions rather than a 'one off' event. To achieve this process of discussion, services need to consider how to trigger conversations, support a wider group of clinicians to lead discussions and protect the time needed to undertake them. A potential trigger to care planning discussion is the use of a care bundle following hospital discharge, as in respiratory care [30]. Public engagement initiatives may also have a role in introducing the concept of frailty more widely, normalising discussions around risk and end of life care and providing opportunities to consider and clarify priorities [31]. This approach was trialled by our research team at an Imperial College engagement event and was well received and we have produced an animation to aide these discussions.

A strength of our study was the use of an online approach to interview people living with frailty and their carers. We purposively sampled for demographic diversity and a range healthcare experiences. By recruiting a broad range of clinicians, we gained an understanding of different professional roles and challenges in primary and secondary care.

Our study has some limitations. Although it provides a broad range of perspectives caution should be taken when considering the transferability of the findings. There were a limited number of male participants -particularly clinicians and carers. Although we gathered some perspectives on the influence of different backgrounds and experiences, this is a complex area requiring continued research. Finally, our study did not involve longitudinal perspectives or access to documented care plans which may have enhanced our insights.

## Conclusion

This qualitative study of patient, carer and clinicians' perspectives found that treatment escalation planning, in frailty, was undermined by negative perceptions around frailty and the numerous uncertainties that can arise during these discussions. In the absence of treatment escalation planning, decision-making during an acute health crisis focussed on survival or risk aversion, which all parties agreed was a poor long-term strategy. We found that strategies for facilitating planning treatment escalation included fostering a shared understanding, careful consideration of timing and the delivery by trusted clinicians supported by a wider multidisciplinary team.

## Acknowledgments

We would like to thank all the people who gave up their time to take part in our study especially as this was during the COVID 19 pandemic.

## Author Contributions

**Conceptualization:** Stephen J. Brett, Helen Ward.

**Data curation:** Adam Lound.

**Formal analysis:** Adam Lound, Jane Bruton, Kathryn Jones, Stephen J. Brett, Helen Ward.

**Funding acquisition:** Stephen J. Brett, Helen Ward.

**Investigation:** Adam Lound, Jane Bruton, Kathryn Jones, Jamie Gross, Stephen J. Brett.

**Methodology:** Adam Lound, Jane Bruton, Kathryn Jones, Stephen J. Brett, Helen Ward.

**Project administration:** Adam Lound.

**Supervision:** Jane Bruton, Kathryn Jones, Stephen J. Brett, Helen Ward.

**Validation:** Adam Lound, Jane Bruton, Kathryn Jones, Nira Shah, Barry Williams, Jamie Gross, Stephen J. Brett, Helen Ward.

**Writing – original draft:** Adam Lound.

**Writing – review & editing:** Jane Bruton, Kathryn Jones, Nira Shah, Barry Williams, Jamie Gross, Benjamin Post, Sophie Day, Stephen J. Brett, Helen Ward.

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
