## [Decision Letter · Decision Letter 0]

28 Jun 2023

PONE-D-23-13727"I'd rather wait and see what's around the corner": A multi-perspective qualitative study of treatment escalation planning in frailtyPLOS ONE

Dear Dr. Jones,

Thank you for submitting your manuscript to PLOS ONE. After careful consideration, we feel that it has merit but does not fully meet PLOS ONE’s publication criteria as it currently stands. Therefore, we invite you to submit a revised version of the manuscript that addresses the points raised during the review process. As a result of the reviewers' and academic editor's decisions, your paper needs further improvement in terms of methodology, data accessibility information in accordance with PLOS One's data policy, and the description of the results and conclusions. Respond appropriately to the reviewers' comments.

We look forward to receiving your revised manuscript.

Kind regards,

Sefonias Getachew, MPH, PhD

Academic Editor

PLOS ONE

4. We note that Figure 1 in your submission contain copyrighted images. All PLOS content is published under the Creative Commons Attribution License (CC BY 4.0), which means that the manuscript, images, and Supporting Information files will be freely available online, and any third party is permitted to access, download, copy, distribute, and use these materials in any way, even commercially, with proper attribution. For more information, see our copyright guidelines: http://journals.plos.org/plosone/s/licenses-and-copyright.

b.If you are unable to obtain permission from the original copyright holder to publish these figures under the CC BY 4.0 license or if the copyright holder’s requirements are incompatible with the CC BY 4.0 license, please either i) remove the figure or ii) supply a replacement figure that complies with the CC BY 4.0 license. Please check copyright information on all replacement figures and update the figure caption with source information. If applicable, please specify in the figure caption text when a figure is similar but not identical to the original image and is therefore for illustrative purposes only.

Reviewers' comments:

Reviewer's Responses to Questions

**Comments to the Author**

1. Is the manuscript technically sound, and do the data support the conclusions?

Reviewer #1: No

Reviewer #2: Yes

2. Has the statistical analysis been performed appropriately and rigorously? 

Reviewer #1: N/A

Reviewer #2: Yes

3. Have the authors made all data underlying the findings in their manuscript fully available?

Reviewer #1: No

Reviewer #2: Yes

4. Is the manuscript presented in an intelligible fashion and written in standard English?

Reviewer #1: Yes

Reviewer #2: Yes

5. Review Comments to the Author

Reviewer #1: This paper presents an important contribution in understanding treatment escalation among persons with frailty. However, as written the methods are not technically sound. The authors need to explain how methods of phenomenology are applied within the framework analysis as the underlying methodologies of phenomenology and framework analysis are incongruent. Phenomenology typically uses unstructured interviews to develop a deep understanding, or essence, of a phenomenon and requires the researcher to carefully bracket personal biases. In contrast, framework analysis is driven by a priori ideas and pre-determined questions. As a result of this lack of clarity on methods, the results are not presented in a manner that is congruent with either method.

The finding presented are rich and meaningful, and I encourage the authors to choose one qualitative method and re-analyze the data.

Reviewer #2: The manuscript is technically sound. Under Abstract section: please try to further explain the Methods part and the Conclusion: Can you clearly state your relevant findings and recommendations? Can you remove the bullet points you used to present the themes? Please write them in sentence form.

6. PLOS authors have the option to publish the peer review history of their article (what does this mean?). If published, this will include your full peer review and any attached files.

Reviewer #1: **Yes: **Priscilla K Gazarian

Reviewer #2: No

---

## [Author Response · Author response to Decision Letter 0]

28 Jul 2023

Please see detailed response in our Response to reviewers letter which is attached.

---

## [Editor Report · Decision Letter 1]

21 Aug 2023

PONE-D-23-13727R1"I'd rather wait and see what's around the corner": A multi-perspective qualitative study of treatment escalation planning in frailtyPLOS ONE

Dear Dr. Lound,

Thank you for submitting your manuscript to PLOS ONE. After careful consideration, we feel that it has merit but does not fully meet PLOS ONE’s publication criteria as it currently stands. Therefore, we invite you to submit a revised version of the manuscript that addresses the points raised during the review process.

Please revise some of the sections in your manuscripts. You are advised to amend again the abstract section and structure it as the introduction, method, result, and concussion sections. Keep the keywords as indicated at the end.  In the main document, again the heading used as finding is to be changed to ‘Result’.  Remove also the Figure 1 included in the document and submit it as a separate file in the system based on instructions for Figure documents. However, you have to state the caption and place where the figure to be kept. It is good to remove also the subheadings stated as main findings, and comparison to other literature, and others in the discussion section.

We look forward to receiving your revised manuscript.

Kind regards,

Sefonias Getachew, MPH, PhD

Academic Editor

PLOS ONE
---

## [Author Response · Author response to Decision Letter 1]

25 Aug 2023

We have made all teh editorial changes suggested as outlined in our response to the editor.

---

## [Editor Report · Decision Letter 2]

10 Sep 2023

"I'd rather wait and see what's around the corner": A multi-perspective qualitative study of treatment escalation planning in frailty

PONE-D-23-13727R2

Dear Dr. Lound,

We’re pleased to inform you that your manuscript has been judged scientifically suitable for publication and will be formally accepted for publication once it meets all outstanding technical requirements.

Kind regards,

Sefonias Getachew, MPH, PhD

Academic Editor

PLOS ONE

---

## [Editor Report · Acceptance letter]

12 Sep 2023

PONE-D-23-13727R2 

“I’d rather wait and see what’s around the corner”:
A multi-perspective qualitative study of treatment escalation planning in frailty 

Dear Dr. Lound:

I'm pleased to inform you that your manuscript has been deemed suitable for publication in PLOS ONE. Congratulations! Your manuscript is now with our production department. 

Kind regards, 

on behalf of

Dr Sefonias Getachew 

Academic Editor

PLOS ONE